# Current Trends in Metal Biomining with a Focus on Genomics Aspects and Attention to Arsenopyrite Leaching—A Review

**DOI:** 10.3390/microorganisms11010186

**Published:** 2023-01-11

**Authors:** Tatiana Abashina, Mikhail Vainshtein

**Affiliations:** Federal Research Center “Pushchino Scientific Center for Biological Research of the Russian Academy of Sciences”, Skryabin Institute of Biochemistry and Physiology of Microorganisms RAS, Prospect Nauki 5, 142290 Pushchino, Russia

**Keywords:** bioleaching, bacteria, archaea, genetic modifications, arsenopyrite

## Abstract

The presented review is based on scientific microbiological articles and patents in the field of biomining valuable metals. The main attention is paid to publications of the last two decades, which illustrate some shifts in objects of interest and modern trends both in general and applied microbiology. The review demonstrates that microbial bioleaching continues to develop actively, despite various problems in its industrial application. The previous classic trends in the microbial bioleaching persist and remain unchanged, including (i) the search for and selection of new effective species and strains and (ii) technical optimization of the bioleaching process. Moreover, new trends were formed during the last decades with an emphasis on the phylogeny of leaching microbiota and on genomes of the leaching microorganisms. This area of genomics provides new, interesting information and forms a basis for the subsequent construction of new leaching strains. For example, this review mentions some changed strains with increased resistance to toxic compounds. Additionally, the review considers some problems of bioleaching valuable metals from toxic arsenopyrite.

## 1. Introduction

For many centuries, the main approach to the industrial recovery of metals was pyrometallurgy, i.e., the smelting of metals from ore at high temperatures. Pyrometallurgy is profitable and economically justified when it is provided with a high metal content in mineral raw materials. In the case of low metal content, pyrometallurgy is less profitable or impossible because (i) it requires large economic costs for heating the mineral ballast, (ii) separation of the molten metal is hindered due to the large mass of the mineral ballast. It has already been known for centuries that various metals can be oxidized and their compounds can be dissolved, especially in acidic aqueous solutions. This process was known, at least, since the Middle Ages by the colored streams flowed out of ore dumps washed with rain: blue (copper) or brown (iron). Thus, the extraction of valuable metals from low-grade minerals materials can be beneficial when using hydrometallurgy. Hydrometallurgy, including bioleaching, allows the leaching of metals from crushed mineral materials in reactors (vat leaching), the washing of ore heaps with the following collection of enriched effluents (heap leaching), and even underground leaching with periodic pumping of saturated solutions (in situ leaching) [1,2,3,4].

Microbiologists have found that a number of microorganisms are capable of oxidizing individual minerals and/or forming acids, leading to the destruction of refractory rocks and extraction of valuable metals. The first patent on the complex process of chemical and biological gold leaching was published in 1958 in the USA [5]. Bioleaching continues to be successful in industry [6,7], and interest in the industrial bioleaching valuable metals continues. An indicator of this interest is the number of claimed patents and submitted patent applications in the United States from 1988 to the present. These data are presented in the US-PGPUB, USPAT, and USOCR databases (https://ppubs.uspto.gov/pubwebapp/, accessed on 28 November 2022) and found by combinations search terms “gold AND bioleaching” and “nickel AND bioleaching AND bacteria” (Figure 1). While patents and patent applications address different processes and aspects of them, the general interest in bioleaching is not decreased: the number of the claimed patents and filings in this area remain roughly constant.

Historically, bioleaching has been evolved primarily as extraction of more expensive metals, primarily gold. As well, at the first stages, it used only acidophilic autotrophic sulfide-oxidizing bacteria and archaea. These approaches are easy to understand because they were selected by their economic success: in the first case, metal mining is more profitable due to the cost of the product, in the second, due to cost reduction. The cost reduction in the latter case is achieved due to the facts that (1) autotrophic bacteria do not require any additional carbon substrates, since they synthesize biomass from CO_2_, (2) acidophiles carry out processes under acidic conditions that favor the metals dissolution, (3) some chemoautotrophic bacteria can use some mineral ferrous and sulfide compounds as energy substrates and thus promote the destruction of the minerals.

Metal biomining and metal bioleaching are very close terms. However, the present review is limited mainly with biomining, which includes microbial metal recovery from ores, dumps, and tails while the area of bioleaching is wider and includes metal recovery from secondary sources, e.g., spent batteries, e-waste etc. [8,9,10,11,12]. Leaching microorganisms in the e-waste bioleaching are often presented with neutrophilic and heterotrophic bacteria.

Table 1 presents a modern list of bacteria and archaea of those genera, whose representatives are capable of oxidizing sulfur and/or ferrous compounds in biomining processes. The formation of sulfuric acid from the sulfide minerals was always discussed as especially promising for industrial bioleaching. The table does not include heterotrophic and non-acidophilic bacteria which decompose silicate minerals, since they include non-specialized bacteria and the mechanisms of silicate destruction are still not clear and debatable [13,14]. Moreover, the application of various heterotrophic microorganisms for metal leaching from silicate mineral material has its disadvantage because these bacteria need organic substrates.

Another general disadvantage of bioleaching is a low rate of the biomining processes at low temperatures. To provide optimal temperature conditions, industrial customers prefer vat leaching for heap leaching because use of the reactors (tanks) permits to provide optimal temperatures. The choice between leaching in a tank or in a heap is determined by the ambient temperature and the economic cost of heating the reactor [7,16].

The last preference can be illustrated with Table 2 and Table 3, which compare the number of patents and patent applications for bioleaching, found using a search on (i) genera names of leaching bacterial/archaeal organisms, (ii) with or without search word “reactor”. The search was carried out separately for the in the US patent databases (Table 2) and the patent database of the Russian Federation (Table 3). The search showed that the US patents and patent applications with the reactor bioleaching formed the dominating group in each genus: 54.2–100% (Table 2). The total number of patents and patent applications in the Russian Federation (Table 3) is less than in the United States. The share of documents on the reactor leaching is lower too. The lowest values (27.5–31.7%) are shown for heterotrophic bacteria *Bacillus* and *Pseudomonas*. Thus, there are two different strategies of industrial bioleaching: predominance of some regulated processes in tanks in the US and predominance of the open mining in the Russian Federation.

It is also interesting to note that the generic name *Acidithiobacillus* appeared relatively recently, after the description of this new genus which had been a part of the former genus *Thiobacillus*, so the patents and applications for this genus only cover the period 2003–2022 and represent the most recent trends. The share of the claimed patents and applications with the keyword “reactor” for *Acidithiobacillus* is more than 90% in the US (Table 2) and only 35.9% in the Russian Federation (Table 3).

A good example of industrial metal recovery is the copper bioleaching, Quebrada Blanca, Chile [17]. Selected type of bioleaching largely determines efficiency of the process. Direct metal bioleaching from the dumps without prior preparation means (1) treatment of material with non-uniform particle size, without mixing, (2) absence of any temperature and aeration control. In this regard, a homogeneous heap is better, while a uniform structure simplifies the process. These are arguments in favor of why it is most common on an industrial scale. Heap bioleaching is widely used in copper production to process copper-poor ores. Maximum control is achieved in stirred reactors where many parameters can be controlled: pH, redox, temperature, and aeration. However, the use of reactors is limited by the financial costs of their operation [18]. Currently worldwide copper production is provided by heap bioleaching. Tank (reactor) bioleaching is not economically feasible in obtaining low-value products, and therefore it is mainly used for the extraction of valuable metals [19,20].

As mentioned above, the metal bioleaching process was claimed with a patent half a century ago. Since this discovery, there were various attempts to intensify metal mining processes by bioleaching methods or to find some very new groups of leaching prokaryotes.

There are a number of recent good reviews on metal mining/bioleaching (where biomining is limited with natural raw materials and bioleaching includes technogenic secondary sources) [1,7,10,19,21,22,23]. Most of them are devoted to the successes in industrial bioleaching. The aim of the present review is neither to supplement these reviews nor to form a digest of the most modern publications on the theme, but to comprise the most stable trends through the last decades. According both to the research publications and to the patent applications, dominating efforts are currently concentrated in the following interesting and promising areas:Phylogeny of leaching microbiota and search for new leaching species/strains, which include:−Study on microbial biodiversity: search for new active species/strains capable of more active processes or capable of carrying out little known processes, or more resistant to toxicants, or stimulated by additives and changing process conditions;−Study on genomes of the leaching bacteria to identify genetically determined resistance to toxicants, the possibility of using stimulating additives, and the relationship to oxidative stress;−Attempts to create genetically engineered leaching strains (in fact, these developments are a logical continuation of two previous).Optimization of bioleaching processes via stimulation of microorganisms. This section presented some efforts which remain constant for decades, namely:−Experimental selection of stimulating additives and changes in cultivation/leaching conditions.The special attention is dedicated to investigations on genomics and bioleaching of arsenopyrite as problems of metal bioleaching from refractory arsenic-containing ores.

In accordance with this statement, the present review takes into account research publications and industrial patents on the indicated topics. The sections below describe research investigations in the listed areas.

## 2. Phylogeny of Leaching Microbiota and Search for New Leaching Species/Strains

Main biotechnological interests are based neither on the taxonomy nor the phylogeny of the leaching microorganisms, but on their efficiency in the metal recovery. Both bacteria and archaea can be used in the biomining industry. Obviously, the following microbial characteristics are of primary importance: (i) fundamental ability of strains to oxidize the components of mineral raw materials (iron, sulfur), (ii) activity of the strains under the process conditions (temperature, pH, aeration, pulp density), (iii) resistance of the strains to inhibitory factors (salinity, concentrations of heavy/toxic compounds). All these factors must be determined in the course of isolation or directed selection of the strains, or during their adaptation.

The current situation regarding the phylogenetic distribution of acidophilic forms capable of metal bioleaching metals is presented in Figure 2.

It is interesting to compare the total scheme of phylogenetic diversity with a natural microbiota of the mine drainage presented below in the Figure 3 and, as well, in special investigations on relation between the key genes involved in the bioleaching processes and phylogeny of these microorganisms. The last one was presented by Y. Xiao et al. [25]: objects of the research were microbiota from the heap leaching and leaching drainage (solution). It was shown that (i) the diversity of the microbiota was not limited with acidophilic leaching bacteria, (ii) both microbial communities were dominated by *Acidithiobacillus* and *Leptospirillum*, and (iii) that some genes (*soxC*, *dsrA*, *mer*, *metc*, *rubisco*, *phn*) have significant positive correlation with the micobiota abilities of pyrite bioleaching, sulfur oxidation, and iron oxidation. These results are very interesting because they present some correlation of the leaching processes not only with, e.g., sulfur oxidation (*soxC*), but also with autotrophic carbon assimilation (*rubisco*), polyphosphate transformation (*php*), and so on. However, the whole correlation of the studied genes with abilities of the microbial communities was not very clear (Table 4).

The isolation and study of new species of bacteria and archaea remain ongoing. Accordingly, in a good relation with this current microbiological search, there are reports about applications of new biological agents of bioleaching. Descriptions of new acidithiobacilli [26], sulfobacilli [27], and archaea [28] can serve as good examples of new leaching microorganisms in the last two decades. Of interest are not only new species, but also new strains of known species, which can be more active [29,30].

There is also some possibility of using non-classical leaching bacteria which are not mentioned in the Table 2. A good example of such is the diazotrophic bacterium *Herbaspirillum* sp. strain GW103 for copper leaching from mine soil or mining waste [31]. This strain was resistant to high concentrations of arsenic, copper, zinc and lead, and its resistance was confirmed by genetic analyses which discovered presence of the *copA* and *copB* genes. The maximum value of copper bioleaching was 66% at 30 °C in 60 h [31].

A separate interesting problem lies in the fundamental strategic decision: should a single strain dominate in the leaching association—the strain in relation to which the technological conditions are optimized—or should the leaching process be carried out by a combined association? In the latter case, industrial microbiologists must decide how stable the association should be: different strains and species can lead the process, complementing each other, or replacing each other in competition, depending on fluctuations or changes in the technological conditions. Microbiological technologists conducted special studies to determine efficiency of chalcopyrite leaching of by *A. ferrooxidans* and *A. thiooxidans* separately and by the mixed culture that included both species [32]. Added iron served as an energy source for *A. ferrooxidans*, and added elemental sulfur was an energy source for *A. thiooxidans*. These experiments have shown that the maximum extraction of copper from chalcopyrite was carried out exactly by the mixed culture [32]. The researchers believed that *A. thiooxidans* played a dual role in these bioleaching processes: they participated in direct leaching and prevented the formation of jarosite, which could obstruct the metal extraction process.

Industrial bioleaching at medium temperatures is based mainly on use of mesophilic bacteria *A. ferrooxidans* which are the most common and classic bioleaching agent. Recent studies showed that natural microbial communities in mining waste contained a wide range of prokaryotes, which were largely preserved under technological conditions of leaching [33]. The studies have shown that microbial diversity of the microbiota associations without classical *A. ferrooxidans* can provide even a relatively higher rate of valuable metals leaching. PCR analyses of microbial communities from the drainage of three mines (Dexing Copper Mine, Qibaoshan Copper Mine, and Yaogangxian Tungsten Mine, China) were performed to evaluate the efficiency of bioleaching by the enrichment cultures [34]. Molecular genetic analyses showed that the dominant microorganisms in these processes were strains belonging to the Proteobacteria group, which included representatives of *Nitrospira*, Acidobacteria, and Actinobacteria. The efficiency of the chalcopyrite bioleaching was 99.5% in 15 days. The authors noted that mixed cultures were more efficient in the bioleaching process than the pure culture of *A. ferrooxidans* [34].

It is assumed that the molecular genetic analysis of the leaching population makes it possible to trace which particular strains were in the association initially and which ones began to prevail when the technological conditions were changed. Initially, such works can be based on the study of the natural leaching microbiota or microbiota of the acid-mine drainage [35]. Initial microbiota of the mine waters from the Shanuch deposit (Kamchatka, Russia) largely included representatives of acidophilic species capable of leaching (Figure 3) [36]. As could be expected, in model experiments, the microbiota significantly changed the ratio of strains depending on the pH shifts or changes in the modelled environment.

In the study of chalcopyrite bioleaching by mixed culture of moderate thermophiles, according to rDNA analyses, bacteria *L. ferriphilum* and *A. caldus* and archaea *F. thermophilum* dominated [37]. The real-time PCR analyses showed that *A. caldus* dominated at the beginning of the process but *L. ferriphilum* dominated at the end of the process. At the same time, the amount of *L. ferriphilum* attached to the solid phase fraction increased faster than in solution, while the distribution of *F. thermophilum* remained relatively stable, and only at the end of the process did their relative content begin to increase in the solution. The identification of such features of the temporal and spatial distribution of microorganisms is impossible without molecular methods of analysis.

Similarly, the suchlike shifts in the composition of the leaching microbiota can be also tracked during reactor bioleaching [38].

In whole, the published data showed that phylogenetic study permits to check shifts in leaching microbiota in situ, in model experiments and in reactors. However, at the present time, these data are not sufficient to select industrial strains.

**Figure 3 microorganisms-11-00186-f003:**
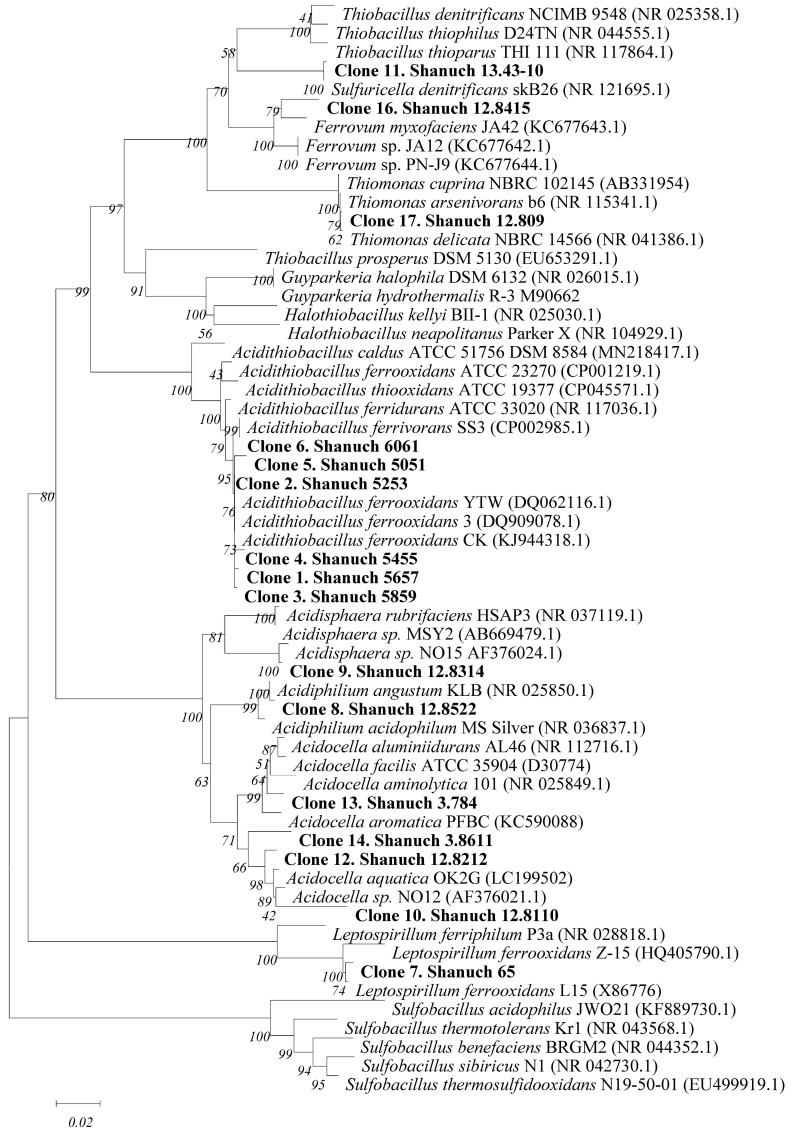
Unrooted phylogenetic tree based on an analysis of the 16S rRNA gene sequences of metal-leaching acidophilic bacteria dominated in the Shanuch deposit microbiota before and after cultivation [36]. The dendrogram was constructed using MEGA 6 software (neighbor-joining analysis) based on a 1536 bp aligned sequence. The sequences obtained in this work are highlighted in bold. Figure 3 is published by permission of Springer Nature Publisher (License Number 5443120489896).

## 3. Optimization of Bioleaching Processes via Stimulation of Microorganisms

For the bioleaching intensification, it is important both to select industrial strains and to increase their activity by correctly changing the technological conditions in accordance with the strain characteristics. These approaches are implemented as inoculation with various strains and technological changes in pH, redox of the medium, aeration, temperature, etc.

### 3.1. Selection of the Leaching Microorganisms

To evaluate leaching activity of various microorganisms, researchers compared strains which belonged to different species and were isolated from acid mine waters [39]. This comparison was carried out on the bioleaching of chalcopyrite in a system that turned out to be preferable: it contained chalcopyrite and, in addition, sulfur and iron as additives. The final output of copper was related both to the amount of iron oxide formed and to the ratio of iron-oxidizing and sulfur-oxidizing bacteria. The latter prevented the formation of iron oxide precipitate (jarosite) that hindered the metals solution [40]. According to the PCR analysis, the initial population contained *A. ferrooxidans* (25.5%), *L. ferrooxidans* (15%), *Acidiphilium* sp. (10.3%), *A. thiooxidans* (16.5%), and *L. ferriphilum* (32.7%). Over time, changes occurred in enriched bacterial associations, leading to an increase in the proportion of acidithiobacilli: during leaching of chalcopyrite with the addition of sulfur, *A. ferrooxidans* and *A. thiooxidans* dominated and reached 43.4 and 39.0%, respectively. In experiments with chalcopyrite without adding sulfur, the proportion of *A. thiooxidans* dropped to 2.5%. Accordingly, the researchers suggested that the decreasing portion of bacteria is non-competitive in bioleaching [39]. In all variants of the experiments, the species *A. ferrooxidans* dominated. On the other hand, there is no doubt that growth of *A. ferrooxidans* was ensured in the experiments by the additional addition of iron and sulfur, both of which were sources of energy for this species, while *A. thiooxidans* oxidized only sulfur, and *L. ferriphilum* only oxidized iron. Anyway, it is also important to note that the maintenance of sulfur-oxidizing bacteria is an additional important industrial factor, since they prevent formation of jarosite, which forms a sedimentary film [40]. The highest ratio of iron-oxidizing bacteria to sulfur-oxidizing bacteria when elemental sulfur was added to chalcopyrite was obtained on the 18th day of bioleaching and amounted to 9:10. In general, the researchers assumed that, under equal conditions, the predominance of iron-oxidizing or sulfur-oxidizing species has to be determined by their initial ratio in the inoculum [39].

It should be noted that the optimal technological parameters of the process must be selected individually for each strain. This fact is an additional argument in favor of the microbiological analysis of which strains of microorganisms are present and which ones are dominant in the industrial leaching process. An example in this regard is the work on optimization of the leaching process with the archaean strain *A. brierleyi*. In work with this species, the bioleaching of concentrate with a high content of molybdenum was studied [41]. The parameters for bioleaching were selected according to the model, which proposed the following parameters: pulp density 0.3%, amount of elemental sulfur 0.2%, the volume of inoculum during inoculation 1%, pH 2.0, temperature 60 °C. The yield of recovered metals under these conditions was 2%, 100%, and 100% for Mo, Cu, and Fe, respectively [41].

The principles of selecting technological parameters when using thermophilic archaea *A. brierleyi* for maximum bioleaching can also be illustrated by another experimental studies [42]. The following factors that determine intensity of bioleaching were chosen as the most significant: (1) pH, (2) pulp density, (3) amount of inoculum, and (4) concentration of added elemental sulfur. Modeling was carried out according to the classic Plaket-Burman factorial model with ten factors. The determined optimal parameters for the maximum extraction of valuable metals by archaea *A. brierleyi* from the processed mineral material were: pH 1.6, solids content in the pulp 0.6%, volume of the applied inoculum 4%, content of the introduced elemental sulfur 4 g/L [42]. The yield of extracted metals from this raw material differed according to the extracted metals but, nevertheless, showed the promise of the selected technological parameters for bioleaching with *A. brierleyi* culture: 35%, 83%, and 69% for Al, Mo, and Ni, respectively [42].

It is interesting that the experimental results obtained by F. Gerayeli et al. [42] are extremely close to the data presented by S.O. Rastegar et al. [41] for the same archaean species. Thus, it can be assumed that the optimal technological parameters could remain constant for species and strains, despite different mineral raw materials.

### 3.2. pH Selection

In general, selection and competitive support of the selected strains is provided due to given technological conditions. Most simply, this mode can be implemented by maintaining the specified values of (1) pH, (2) temperature, and (3) redox in the medium.

It should be taken into account that establishment of certain technological conditions for the bioleaching process plays a dual role: they not only determine the rate of reactions, but also change the composition of the leaching initial population of microorganisms in favor of some strains/species and to the detriment of others. So, A.-K. Halinen et al. [43] and J. He et al. [44] considered the effect of temperature at the constant pH. Change in the level of acidity was of no less importance, since it determines the possibility of the survival and functioning of acidophiles. Researchers [43,45] tested the effect of pH in their leaching experiments with the same Finnish minerals they had used to study the effect of temperature. The oxidation of iron and sulfur increased with decreasing pH. At pH 1.5, 59% Ni, 59% Zn, 13% Cu, and 16% Co were extracted in 140 days. For comparison: at pH 3.0, only 15% Ni, 10% Zn, 5% Cu, and only 0.5% Co were leached over the same time. Accordingly, to stimulate the process, the authors recommended pH 2.0 for heap leaching of this raw material. Interestingly, in all pH variants, according to genetic analysis, the same species dominated: *A. ferrooxidans* and *L. ferrooxidans*, while *S. thermotolerans*, *A. caldus*, *A. thiooxidans*, and other species not identified by the authors remained in the minority [43,45].

As far as microbiologists working in biomining use acidophilic microorganisms, researchers tested a strategy in pH selection to improve bioleaching in a reactor where ferrous iron and elemental sulfur were added as energy substrates [46]. Bioleaching of chalcopyrite was carried out by bacteria of the genus *Acidithiobacillus*. The pH values changed sequentially in the following order: 1.3 → 1.0 → 0.7. The adaptation phase for the strain was reduced from eight to four days, and passivation due to the formation of jarosite significantly decreased. The final production of copper in solution reached 89.1 mg/L (at an average rate of 2.2 mg/L-day), which exceeded the parameters for the parallel blank bioleaching experiment (blank version was a bioleaching process without any additional pH regulation) by 52.8% [46].

### 3.3. Effects of Redox and Temperature Conditions

Redox value can regulate activity of both chemical and biochemical oxidation processes. It was shown that the redox values had a greater effect on the rate of pyrite leaching than the number and condition of the introduced bacteria [47,48]. Thus, the redox regulation is a mandatory and important factor for the regulation and stimulation of bioleaching processes. In practice, it is usually provided by aeration (dissolved oxygen as an oxidizing agent) or by a ratio of ferrous and ferric compounds if their concentrations are high.

Most mining wastes and dumps containing copper belong to the group of sulfide minerals, such as chalcopyrite (CuFeS_2_), from which the extraction of copper is difficult due to passivation of the mineral surface. It has already been mentioned above that the released and oxidized iron, which forms jarosite, takes part in the formation of such a film. The electrochemical modeling studies carried out in the presence and in the absence of leaching microorganisms showed that the deposition of jarosite as a passivating film strongly depends on the redox value [49]. In this case, sulfur-oxidizing bacteria can participate in the partial regeneration of oxidized iron, preventing the formation of jarosite. Thus, in these cases, the redox role is not only to ensure the possibility of iron oxidation by iron-oxidizing microorganisms, but also to prevent the mineral passivation by supporting sulfur-oxidizing bacteria.

It was shown that during chalcopyrite leaching by *Acidiphilium* redox increased from approximately 600 mV up to about 900 mV [50]. These data are in a good accordance with suggestion to regulate the bioleaching process with redox of the medium [51].

In case of establishing a predetermined temperature for the process, the microbial selection usually takes place at the level of species. During the long history of the metal biomining, researchers compared microorganisms at temperatures from low to high ones. A very good example is comparison of temperatures from 7 to 50 °C during bioleaching of shale raw materials with a low content of various metals where the main valuable mineral was pentlandite [45]. The used microbial leaching association contained iron- and sulfur-oxidizing microorganisms and was obtained from a mine drainage; leaching was carried out in column-type reactors. At 7 and 21 °C, redox value in the leaching solution was 500–600 mV while at 35 and 50 °C it varied from 300 to 500 mV. The microbial oxidation of iron at 7 °C started after a 20-day lag-phase. After 60 days of the process, the total iron content (Fe_total_) and the ratio of Fe_total_ to Fe^2+^ were higher in the column at 7 °C than at other temperatures. At 50 °C during the same time, all dissolved iron remained in the ferrous form, i.e., no microbial oxidation occurred. The highest leaching of valuable metals (26% Ni, 18% Zn, 6% Co) was obtained in 140 days at 21 °C. At 50 °C, bioleaching was minimal due to the lack of formation of iron oxide as an oxidizing agent. According to the PCR analyses, microbial composition depended on the temperature regime: *A. ferrooxidans* was the dominant species in the leaching solution at 7 °C while *L. ferrooxidans* dominated at 35 °C and *Sulfolobus thermotolerans* dominated at 50 °C. In whole, the number of microorganisms in the liquid phase was higher at 7 °C. Interestingly, the enrichment culture was obtained from the northern region of Finland, which may explain the predominant activity of bacteria at low temperatures, although it does not explain the presence and activity of thermophilic archaea *S. thermotolerans* [44].

Studies conducted using molecular genetic analysis may detect changes in the microbial leaching community of the reactor. An example of such technique for checking possible changes in the leaching community associated with the temperature regime is the work of J. He et al. [44]. In this work, strains of the species *A. ferrooxidans*, *A. thiooxidans*, *A. caldus*, *L. ferrooxidans*, and *S. thermosulfidooxidans* were analyzed at different temperatures and various initial pH values during pyrite bioleaching. The PCR analysis of 16S rDNA showed that the dominance of certain species during pyrite leaching completely changed together with a change in temperature at a constant pH value. *A. ferrooxidans* and *A. thiooxidans* dominated at 30–35 °C; *L. ferrooxidans* was found at 35 °C, and at 40 °C it was among the dominant ones along with *A. caldus* and *S. thermosulfidooxidans*. The latter species was dominant at 40–45 °C. The authors concluded that temperature was the main factor determining the composition of the microbial community during pyrite leaching [44].

Temperature control seems to be a convenient approach to control the bioleaching process, since in this case the temperature parameters establish a kind of selection for active thermophilic microorganisms. Moreover, the rate of abiogenic chemical reactions increases with increasing temperature. The oxidation of mineral sulfide concentrates by thermophilic acidophilic archaea was described as early as the 1970s when all thermophilic archaea with unclear cell morphology were presumably assigned to the genus *Sulfolobus*. Later, a number of these strains were described as *Acidianus* and *Metallosphaera*, and a microbial community dominated by *A. brierleyi* was used in bioreactors at 70 °C. Microbial communities used in bioleaching in reactors at high temperatures are still poorly characterized, although these organisms are thought to be thermophilic acidophilic archaea [52]. The classic species used in bioleaching at high temperatures remains *S. metallicus* [53]. Along with this, it must be noted that maintaining a strict regime of high temperatures is economically costly and technically complicated.

### 3.4. Stimulation with Organic Supplements

As it was mentioned above through text of the review, the traditional leaching microorganisms are acidophilic autotrophs. In various areas of biotechnology where heterotrophic microorganisms are used, microbial activity is commonly stimulated with addition of some organic substrates. Autotrophy greatly limits the possibility of stimulation in this way. However, an interesting advance in this area is the attempt to stimulate autotrophs with very specific organic compounds. For example, the US Patent No. 8,728,785 [54] states that the addition of lycananthase lipoprotein during leaching by *A. thiooxidans* or *A. ferrooxidans* bacteria increased the yield of copper up to 20%. According to the patent CN 110,669,937 [55], the addition of glutathione increased the rate of metal leaching by acidophilus bacteria too.

Three decades ago, J. Pronk with et al. [56] already showed that the growth of autotrophic bacteria *A. ferrooxidans* (“*Thiobacillus ferrooxidans*”) can be stimulated with formic acid. According to the published data, formic acid was not used as a growth substrate but provided some energy requirements. In addition, these authors also proposed pre-cultivation of *A. ferrooxidans* on formate for subsequent leaching of metals from ores and patented this approach [57].

Recently, it was shown that formate stimulated the neutrophilic autotrophic bacteria *G. halophila* [58] which are neutrophilic autotrophs able of oxidizing sulfide and thiosulfate to sulfuric acid at an optimum pH of 7.0–7.3 [59]. The use of neutrophilic autotrophic thionic bacteria for formate-stimulated leaching could be an environmentally friendly technology as it reduces the consumption of sulfuric acid and simplifies the subsequent disposal of acid waste. Formate stimulation of the growth and leaching of sulfide ore in *G. halophila* may indicate that such additive is promising for autotrophs.

At the same time, a new attempt to use formate to stimulate *A. ferrooxidans* showed an increase in biomass in the culture and no increase in the leaching of the arsenopyrite concentrate [60]. It can be assumed that bacteria switched to the use of formate as a more accessible energy substrate than oxidizing refractory arsenopyrite.

## 4. Problems of Bioleaching of Metals from Refractory Arsenic-Containing Ores

Optimization of metals bioleaching is a complex task, solved in different cases by different methods in accordance with the treated mineral raw materials and target extracted metals. One of the most difficult problems in this area is the extraction of valuable metals from the refractory arsenopyrite ores. Arsenopyrite is a common mineral with an arsenic content of up to 45% [61]; it often associates gold [62,63,64] as inclusions in the crystal structure [63]. Accordingly, the gold recovery by leaching is very low and arsenopyrite must be degraded before the gold can be recovered. Acid mine drainage of arsenopyrite is one of the most serious environmental problems [64,65,66], as it is toxic due to arsenic contamination [62].

Industrial application of leaching microorganisms is due in mine waters or pulp, or artificial leaching solutions where dissolved toxic chemical compounds can reach high concentrations. Thus, both toxicity of the mineral ore material and resistance of the leaching bacteria have a great significance for biomining. There are various points of view on the problem of the arsenopyrite leaching which include not only toxicity, but also the formation of scorodite and solubility of the minerals surface [67,68].

Bioleaching of arsenopyrite is an important technology due to its environmental and economic benefits [69,70,71], it is applied in many countries. Meanwhile, countries which import this mineral concentrate are expected to impose economic penalties based on the arsenic content of the concentrate [72,73]. Thus, from economical point of view, it is helpful to leach arsenic before the export of concentrate. Since arsenic is a highly toxic element, it also significantly affects the growth and metabolic functions of various leaching microorganisms. To increase the efficiency of the arsenopyrite biooxidation, the selected leaching strains must have a mechanism of resistance to As [72]. Table 5 lists some of the arsenic-resistant strains that have been adapted in the course of experimental investigations. Adaptation work can be excluded if strains are preselected for the presence of the arsenic resistance genes [60,71].

Preliminary selection of the bioleaching agents has a great significance because researches can choose more As-resistant and more active microorganisms. So, studies have shown that thermophilic microorganisms dissolves enargit better than mesophilic ones [78]. B. Escobar et al. [79] used *Sulfolobus* sp. BC for leaching enargite at 70 °C: in the absence of Fe^3+^, *Sulfolobus* sp. BC oxidized enargite by contact during the cell sorption on the mineral surface. J. Song et al. [80] proposed a model for the enargite bioleaching with two pathways: (i) contact leaching when cells are placed directly on the surface of mineral particles and (ii) non-contact leaching when dissolved biogenic products act as intermediates in the solution. According to this model, the proportion of non-contact leaching in this model was up to 95.5%: AsO^2−^ was oxidized with Fe^3+^ to AsO_4_^3−^ and caused precipitation of iron arsenate. Although arsenic is toxic to microorganisms, their formation of iron arsenate reduced concentration of arsenic in the leaching solution and, accordingly, significantly reduces As toxicity [79,80]. K. Takatsugi et al. [81] studied the enargite bioleaching with *A. brierleyi* and showed that enargite oxidation was mediated by Fe^3+^. After leaching at 70 °C for 27 days, the copper recovery reached 90.9%, while arsenic recovery was only 5.9% [81]. During the bioleaching of arsenopyrite, arsenic dissolved in the form of arsenate, but a passive film was formed, consisting mainly of jarosite, elemental sulfur, and iron arsenate on the surface of the mineral, and thus suppressed the efficiency of this bioleaching [82,83].

Since arsenopyrite contains iron, addition of iron oxide Fe^3+^ as a supplementing oxidizing agent is sometimes not essential in the bioleaching process, since Fe^3+^ is formed during the oxidation of arsenopyrite ferrous iron by microorganisms. During the biooxidation of arsenopyrite by *L. ferrooxidans*, the surface of arsenopyrite was covered with an extracellular polymeric substance, and the surface corrosion of minerals, detected by electron microscopy, suggested a contact leaching mechanism [84]. D.R. Zhang et al. [85] studied the mechanism of biooxidation of arsenopyrite by *S. thermosulfidooxidans* with or without Fe^3+^ and showed that, when Fe^3+^ was added, the surface of the mineral was mainly covered with loose structure sediments consisting of jarosite and iron arsenate. Meanwhile, without the addition of Fe^3+^, the surface of the mineral was mainly covered with sediments with a dense structure, consisting mainly of jarosite, iron arsenate, and elemental sulfur.

Attempts to solve various problems associated with the bioleaching of arsenopyrite are illustrated by the patents of recent years on this topic claimed in the Russian Federation (Table 6).

Table 6 shows that in Russia, in the field of bioleaching of refractory arsenopyrite ores and concentrates, the main patentable innovations were devoted to the oxidation regime and the selection of effective strains.

There are at least three families of the *ars* (arsenic resistance system) gene group which can be distinguished by their protein structures, reduction mechanisms, and location of catalytic cysteine residues. According to analysis of bacterial genomes, a large number of phylogenetically diverse prokaryotes are able to transform As(V) and As(III) under a wide range of environmental conditions [92,93,94]. For example, genetic analyzes of the *A. ferrooxidans* VKM B-3655, originally isolated from an arsenopyrite deposit, revealed the presence of the *ars* [60]. The *arsC* gene encodes arsene reductase and is involved in the transformation of As(V) into As(III), which is then excreted by the *arsB* arsenite pump. The strain *A. ferrooxidans* VKM B-3655 also possesses two thyredoxin-dependent arsene reductases [60]. In addition, it also had the *arsM* genes encoding putative arsenite methyltransferases.

The whole scheme of the genome revealed presence of the *ars* gene group in the various *A. ferrooxidans* strains is shown in the Figure 4. The *arsC* gene encodes arsenate reductase and is involved in the transformation of As(V) into As(III), which is then excreted by the *arsB* arsenite pump. This mechanism provides arsenic resistance for bacteria, although it increases toxicity of the surrounding medium. It was discovered [60] that the studied strain had all genetic determinants for the operation of the arsenic resistance system in two variants. In addition, it also had the *arsM* genes encoding putative arsenite methyltransferases. As shown by metagenomic studies, the *arsM* and *arsI* genes are often found in representatives of different species of the same community, forming an arsenate biogeochemical cycle in nature [95].

## 5. Genetic Modifications of Leaching Microorganisms

Studies on genomes of leaching bacteria to reveal their resistance to toxicants, including investigations described in the previous section [60,96], represent a step towards the genetic modification of strains.

Such modifications can be carried out by introducing a plasmid, e.g., to increase resistance to high concentrations of arsenic [75,97] or to increase resistance to nickel [98].

Accordingly, the next interesting modern direction is some genetic engineering of improved strains. Development of this trend in the area of bioleaching was initiated by studying the genomics of bioleaching microorganisms. This was accompanied by an accumulation of data over the past decades [89,90,91,92,93,94,95,96,97,98,99,100]. The present time seems to mark the beginning of new stage, namely the genetic selection of bacteria with desired properties. To speculate, more and more genomic data appear. Thus, industrial researchers can try to select a strain for a specific task/ore.

In addition to the studies on adaptation and increase in resistance mentioned above, an interesting work on enhancement of *M. sedula* bioleaching should be mentioned [101]. In this work, laboratory adaptation and selection of the archaea *M. sedula* were accompanied with (1) genetic mutation analyses and (2) transcriptomics to study stress-induced gene expression.

## 6. Concluding Remarks

Biomining and bioleaching of valuable metals from mineral raw materials are combined chemical and biological processes in which microorganisms are used in catalyzing the metal oxidation and to change the structure of the treated minerals, change the pH and redox in the medium, and to destroy refractory ores. The economic limitations of biomining for metal recovery are inefficient productivity at low temperatures [102] and the need for the additional biomass preparation. At the same time, the use of microorganisms in hydrometallurgy is expedient in warm climatic zones and during warm season in a moderate climate. A comparison of the US and the Russian Federation patents showed that the first prefer regulated reactors while the second used more heap leaching in situ. Anyway, the use of microorganisms is much cheaper than any energy-consuming chemical autoclave leaching. The use of microorganisms in leaching can also reduce introduction of sulfuric acid and, consequently, decrease environmental damage, costs for planting, and environmental restoration [58].

Thus, a significant and important problem arises, namely the search for ways to increase the efficiency of the metal bioleaching. Apparently, bioleaching at temperatures close to the freezing point of aqueous solutions will remain ineffective. However, there are different ways to increase the efficiency of bioleaching at moderate and high temperatures [103]. As follows from the presented review, these approaches are:(1)Permanent and continuous ones:−studying the diversity of microorganisms to search new groups, species, strains capable of bioleaching processes in a wider range of conditions,−the development of bioleaching technologies, including a wide range of approaches: application of the species/strain associations, innovations in conditions and stages of the bioleaching process, use of stimulating additives;(2)New ones which expand with time:−studying genomes of the leaching microbiota to understand interactions inside of the leaching populations,−studying genomes of the leaching bacteria to identify their genetically determined resistance to toxicants, identifying some possible new properties and trying to create genetically engineered leaching strains.

The presented collected data suggest that both the traditional search for new industrial strains and process optimization will continue. Along with this, the genetic study of leaching microbiota will be developed. This will make possible to study the interaction of various species/strains and their replacement. A separate promising area is the study of the genomes of leaching microorganisms for bacterial resistance to toxic compounds. A possible consequence of such work is the genetic engineering of more resistant microorganisms.

In general, this review presents a summary of data over the past two decades illustrating the evolution of bioleaching. It is shown that industrial applications of chemical leaching and bioleaching (chemical-biological leaching) are not mutually exclusive and, conversely, can complement each other.

## Figures and Tables

**Figure 1 microorganisms-11-00186-f001:**
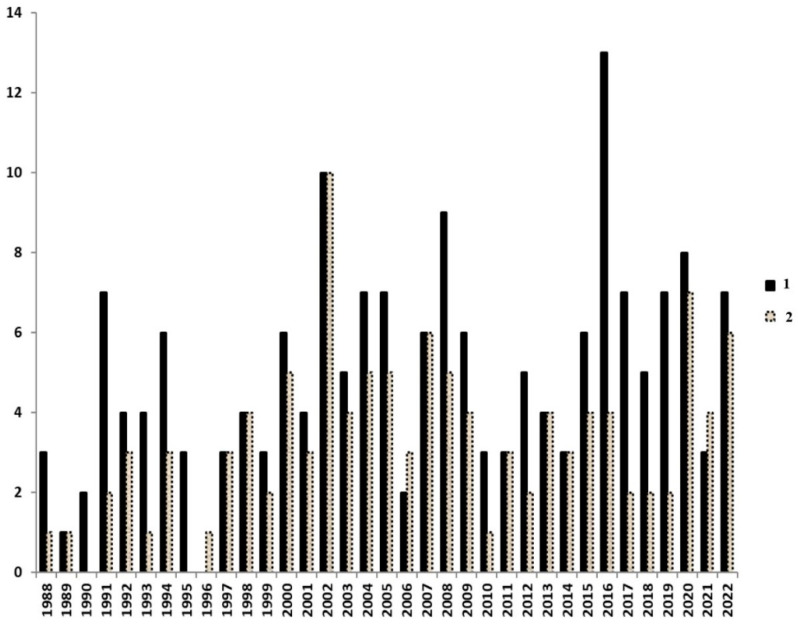
The number of patents and patent applications filed in the US in 1988–2022 according to the US-PGPUB, USPAT, USOCR databases. Search by combinations of search words 1—“gold AND bioleaching” and 2—“nickel AND bioleaching AND bacteria”.

**Figure 2 microorganisms-11-00186-f002:**
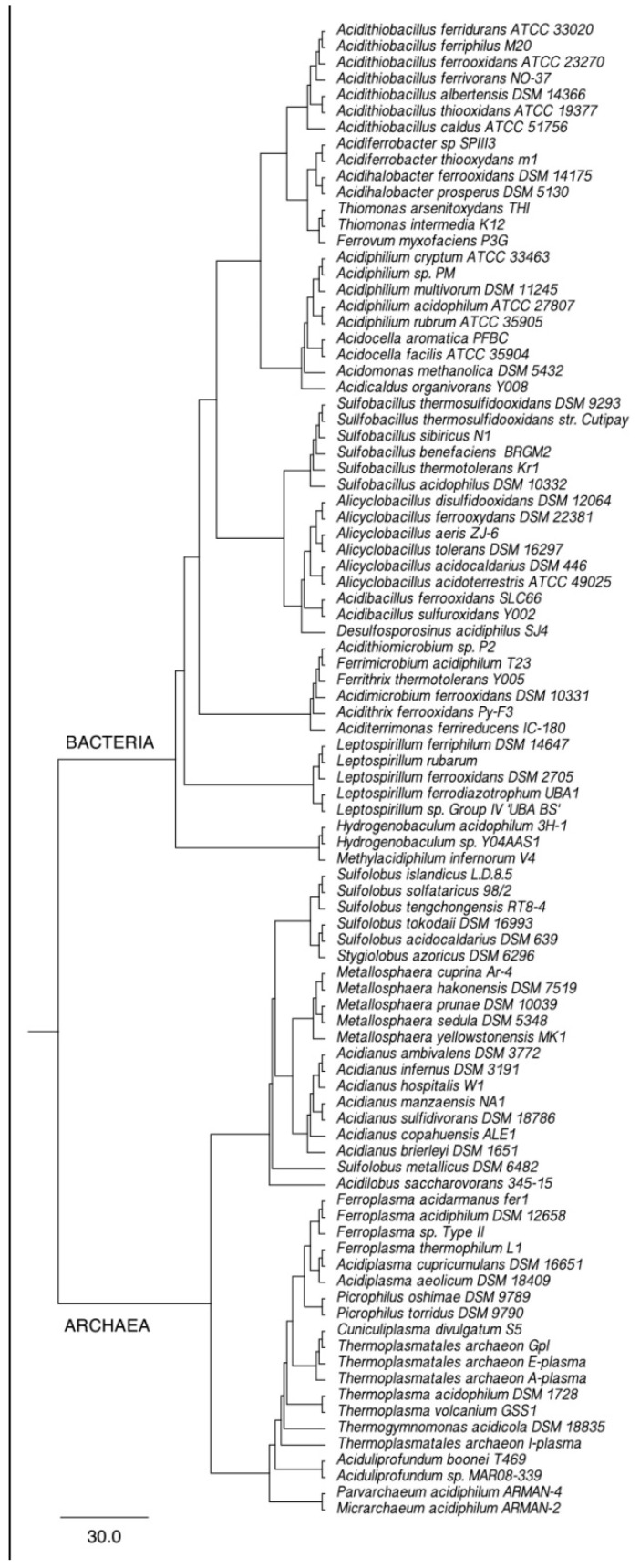
General scheme of phylogenetic diversity in the main groups of acidophilic bacteria and archaea capable of leaching metals [24] (open access).

**Figure 4 microorganisms-11-00186-f004:**
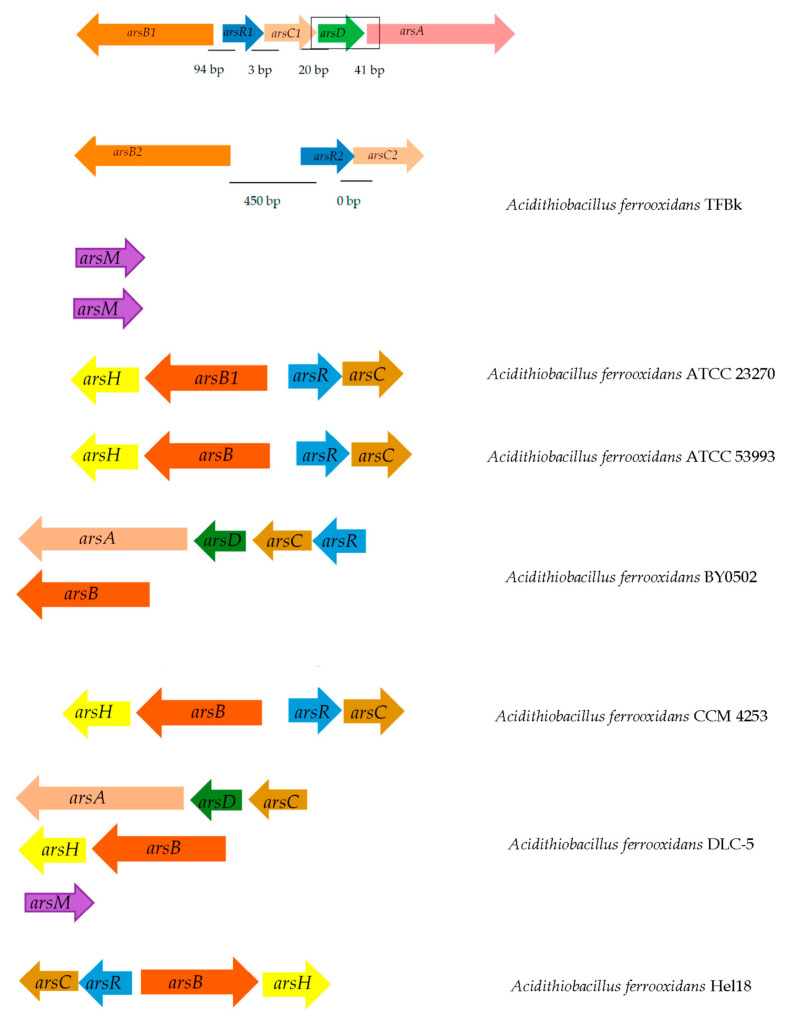
Location of arsenic resistance genes on the chromosome in various strains of *A. ferrooxidans* [60] (open access). The distance between genes is shown as bp scale. Genes encoding: *arsB*, arsenite/antimonite: H^+^ antiporter; *arsR*, transcriptional repressor of the arsenic resistance operon; *arsC*, arsenate reductase; *arsD*, transcriptional repressor and arsenic metal chaperone.

**Table 1 microorganisms-11-00186-t001:** Archaeal (a) and bacterial (b) species of the genera whose representatives were used for the acid metals bioleaching [15].

Modern Name	Former Name	Phylum	pH Optimum	Temperature Optimum, °C
*Acidianus ambivalens*	*Desulfurolobus* *ambivalens*	a	2.5	80
*Acidanus brierley*	*Sulfolobus brierleyi*	a	1.5–2.5	70
*Acidianus infernus*		a	2.5	80
*Acidianus sulfidivorans*		a	2.0	70
*Idiferrobacter thiooxidans*		b	2.2	30
*Acidiphilium acidophilum*	*Thiobacillus acidophilus*	b	4.5	30
*Acidiphilium angustum*		b	2.5–3.0	20
*Acidiphilium cryptum*		b	3.0	28
*Acidiphilium multivorum*		b	3.0	30
*Acidiphilium organovorum*		b	3.5	37
*Acidiphilium rubrum*		b	2.5–3.0	20
*Acidithiobacillus albertensis*	*Thiobacillus albertis*	b	4.4	30
*Acidithiobacillus caldus*	*Thiobacillus caldus*	b	2.5	45
*Acidithiobacillus ferrianus*		b		
*Acidithiobacillus ferridurans*		b	1.8	30
*Acidithiobacillus ferriphilus*		b	2.0	30
*Acidithiobacillus ferrivorans*		b	1.8	25
*Acidithiobacillus ferrooxidans*	*Thiobacillus ferrooxidans*	b	1.8	25
*Acidithiobacillus sulfuriphilus*		b	3.0	25–28
*Acidithiobacillus thiooxidans*	*Thiobacillus thiooxidans,* *Thiobacillus* *concretivorus*	b	4.4	25
*Acidocella aluminiidurans*		b	3.0	37
*Acidocella aminolytica*	*Acidiphilium* *aminilyticum*	b	2.5–3.0	30
*Acidocella facilis*	*Acidiphilium facile*	b	3.0	20
*Ferroplasma acidiphilum*		a	1.6–1.8	35
*Ferroplasma cupricumulans*		a	1.0–1.2	53
*Guyparkeria halophila*	*Halothiobacillus halophilus*	b	7.3–7.5	30
*Guyparkeria hydrothermalis*	*Halothiobacillus hydrothermalis*	b	7.5	35
*Halothiobacillus kellyi*		b	6.6–7.0	37
*Halothiobacillus neapolitanus*	*Thiobacillus neapolitanus*	b	6.6–7.0	25
*Leptospirillum ferriphilum*		b	1.8	37
*Leptospirillum ferrooxidans*		b	1.8	30
*Leptospirillum thermoferrooxidans*		b	3.5	65
*Metallosphaera cuprina*		a	3.0	65
*Metallosphaera hakonensis*	*Sulfolobus hakonensis*	a	3.0	75
*Metallosphaera prunae*		a	3.0	65
*Metallosphaera sedula*		a	1.6–1.8	35
*Sulfobacillus acidophilus*		b	2.0	45
*Sulfobacillus benefaciens*		b	2.0	37
*Sulfobacillus sibiricus*		b	1.9–2.4	50
*Sulfobacillus thermosulfidooxidans*		b	1.9–2.4	50
*Sulfobacillus thermotolerans*		b	1.9–2.4	40
*Sulfolobus acidocaldarius*		a	2.0	70
*Sulfolobus metallicus*		a	2.0	70
*Sulfolobus shibatae*		a	3.0–4.0	75
*Sulfolobus solfataricus*		a	4.0–4.2	70
*Sulfolobus tokodaii*		a	2.0	75
*Sulfolobus yangmingensis*		a	4.0	80
*Thermithiobacillus tepidarius*	*Thiobacillus tepidarius*	b	6.9	43
*Thiobacillus aquaesulis*		b	7.6	42
*Thiobacillus denitrificans*		b	7.0	30
*Thiobacillus prosperus*		b	7.0	35
*Thiobacillus thioparus*		b	6.6	26
*Thiobacillus thiophilus*		b	7.0	25

**Table 2 microorganisms-11-00186-t002:** Number of the US patents and patent applications (1988–2022) found by keywords “bioleaching” or “bioleaching AND reactor” together with the generic names of microorganisms in the US patent databases US-PGPUB, USPAT, USOCR.

Genera	Number of Patents and Patent Applications		
	1. Bioleaching	2. Bioleaching AND Reactor	2:1, %
*Thiobacillus* (w/o *Acidithiobacillus*)	131	81	61.8
*Sulfolobus*	83	53	63.8
*Leptospirillum*	77	52	67.5
*Pseudomonas*	67	43	54.2
*Bacillus*	66	36	54.5
*Sulfobacillus*	55	34	61.8
*Acidithiobacillus* (w/o *Thiobacillus*)	54	49	90.7
*Acidianus*	44	28	63.6
*Thiobacillus* & *Acidithiobacillus*	32	23	71.9
*Thiomicrospira*	5	5	100.0
*Halothiobacillus*	5	4	80.0
*Sulfurimonas*	1	1	100.0

**Table 3 microorganisms-11-00186-t003:** Number of the Russian patents and patent applications (1991–2022) found by keywords “bioleaching” or “bioleaching AND reactor” together with the generic names of microorganisms in the Russian Federal patent database FIPS (https://www.fips.ru/iiss/search.xhtml, accessed on 28 November 2022).

Genera	Number of Patents and Patent Applications		
	1. Bioleaching	2. Bioleaching AND Reactor	2:1, %
*Thiobacillus* (w/o *Acidithiobacillus*)	59	21	35.6
*Sulfolobus*	16	11	68.8
*Leptospirillum*	28	16	57.1
*Pseudomonas*	41	13	31.7
*Bacillus*	40	11	27.5
*Sulfobacillus*	19	10	52.6
*Acidithiobacillus* (w/o *Thiobacillus*)	39	14	35.9
*Acidianus*	11	6	54.5
*Thiobacillus* & *Acidithiobacillus*	8	5	62.5
*Thiomicrospira*	1	1	100.0
*Halothiobacillus*	0	0	-
*Sulfurimonas*	0	0	-

**Table 4 microorganisms-11-00186-t004:** Summary of relationships between the dominant genera and functions by Pearson correlation tests [25] (open access). Significant differences are indicated in bold.

Genus	Ability of Pyrite Bioleaching		Ability of Sulfur Oxidation		Ability of Iron Oxidation	
	r	*p*	r	*p*	r	*p*
Unclassified	0.817	**0.000**	0.482	**0.037**	0.775	**0.000**
*Methylophilus*	−0.664	**0.002**	−0.493	**0.032**	−0.408	0.083
*Methylotenera*	−0.613	**0.005**	−0.414	0.078	−0.386	0.102
*Acidisoma*	−0.309	0.198	−0.031	0.901	−0.340	0.154
*Thiomonas*	0.663	**0.002**	0.342	0.152	0.330	0.168
*Thermogymnomonas*	0.499	**0.030**	0.297	0.218	0.325	0.175
*Leptospirillum*	−0.420	0.074	−0.401	0.089	−0.312	0.193
*Acidithiobacillus*	−0.419	0.074	−0.153	0.533	−0.305	0.204
*Acidiphilium*	−0.312	0.194	0.001	0.997	−0.261	0.280
*Thiobacillus*	0.536	**0.018**	0.235	0.333	0.239	0.324
*Frateuria*	0.421	0.073	0.315	0.189	0.190	0.436
*Hydrotalea*	0.572	**0.011**	0.297	0.216	0.154	0.530

**Table 5 microorganisms-11-00186-t005:** Adaptation of leaching bacteria to arsenic.

Bacteria	Reached As-Resistance, g As/L	Method for Adaptation	References
Mixture of *A. ferrooxidans* and *L. ferrooxidans*	18.0	Culturing in the medium with increasing As-concentration	[74]
*A. caldus*	3.375	Transfer of the plasmida pSDRA4 coding As-resistance	[75]
*L. ferriphilum* YSK	30.0	Culturing in the As-concentration gradient	[76]
*A. thiooxidans* A01	60	Culturing in the As-concentration gradient	[76]
*A. ferrooxidans*	12	Continuous replacing of dominants	[77]

**Table 6 microorganisms-11-00186-t006:** The Russian Federation patents on the arsenopyrite bioleaching (2008–2021).

Title	Main Content	References
Association of microorganisms *Sulfobacillus olimpiadicus*, *Ferroplasma acidiphilum*, *Leptospirillum ferrooxidans* for the oxidation of sulfide gold-bearing concentrate	Preselection of the most effective strains	[86]
Method of bacterial oxidation of gold-bearing sulfide concentrates in the production of gold	3-stages of biooxidation under different conditions	[87]
Method for bacterial oxidation of sulfide gold-bearing concentrates for the gold recovery	4-stages of biooxidation with the different regimes of aeration	[88]
Method for controlling the process of biooxidation of sulfide concentrates	Controlling via aeration for the biooxidation	[89]
Method for bioleaching of refractory gold-bearing sulfide flotation concentrates	Bioleaching is controlled by redox of medium	[51]
*Thermithiobacillus tepidarius* strain for post-oxidation of elemental sulfur in residues of biooxidation of sulfide gold-arsenic concentrate	New industrial strain *Thermithiobacillus tepidarius* OL2018-8.	[90]
Association of microorganisms *Acidithiobacillus thiooxidans*, *Acidiphilum criptum*, *Leptospirillum ferriphillum*, *Ferroplasma acidiphilum* for oxidation of sulfide gold-bearing concentrate	An effective association of the leaching strains	[91]

## Data Availability

The data that support the findings of this review are openly available.

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
