# Peer review of "Current Trends in Metal Biomining with a Focus on Genomics Aspects and Attention to Arsenopyrite Leaching—A Review"

_microorganisms, 2023, doi:10.3390/microorganisms11010186_

Round 1

Reviewer 1 Report

The document is clear in structure and is based on current bibliography. I suggest improving the complement of Table 2 and Figure 2 with other authors. In addition, it is also important to point out successful cases of copper bioleaching, Quebrada Blanca (Chile) for example.

On line 37, please insert a reference that justifies the idea.

On line 65, please check English. Is not clear.

Between lines 74 and 80, please insert a reference that justifies the idea

Figure 2, was made only by reference 6?. What is then the novelty of the review? This comment also applies to Table 2.

Line 245, I suggest g/L

Line 256, do you mean the solution potential? mix potential?

Line 385, but the problem is the non-stabilization of the solution (like scorodite, for example) rather than the reactivity (this is a personal appreciation)

Line 406, please modify the way to cite the author (use the et al)

I recommend complementing the bibliography with an industrial case study https://doi.org/10.3390/min12040487

Author Response

Dear Reviewer,  

First of all, we want to thank for your comments and suggestions which were really helpful.  

Comments are addressed below. The responses to all comments and questions are presented below in the order of your comments.Each comment is first recalled then the corresponding replies are given. The questions and comments of the reviewer are presented below with our answers provided as “point by point”. We corrected and supplemented the manuscript according to all these comments and questions. 

Kindly find the revised editorial article microorganisms-2096604 "Review - Current trends in metal biomining and problems of applied genomics and arsenopyrite leaching". Changes to the text are highlighted in color.  

Comments and Suggestions for Authors

Reviewer 1.

  1. “The document is clear in structure and is based on current bibliography. I suggest improving the complement of Table 2 and Figure 2 with other authors. In addition, it is also important to point out successful cases of copper bioleaching, Quebrada Blanca (Chile) for example.”

-Table 2 and Fig. 2 are supplemented with Tables 3 and 4 and with new relevant additional text. - Reference for the case [Gentina, J.C.; Acevedo, F. Copper Bioleaching in Chile] is included with the appropriate text.

  1. “On line 37, please insert a reference that justifies the idea.”

- The section "Introduction" is supplemented with the appropriate references and relevant text.

  1. “On line 65, please check English. Is not clear.”

- Thank you, the text is improved (the paragraph below Fig. 1).

  1. “Between lines 74 and 80, please insert a reference that justifies the idea.”

- The text is revised, new reference is added (the text below Table 1).

  1. “Figure 2, was made only by reference 6? What is then the novelty of the review? This comment also applies to Table 2.”

-Table 2 and Fig. 2 are supplemented with Tables 3 and 4 and with new relevant additional text.

  1. “Line 245, I suggest g/L.”

- g/l changed for g/L through the whole text.

  1. “Line 256, do you mean the solution potential? mix potential?”

- “Redox potential” is replaced with “redox in the medium” through the whole text.

  1. “Line 385, but the problem is the non-stabilization of the solution (like scorodite, for example) rather than the reactivity (this is a personal appreciation).”

- This point of view and related references are added (page 15, paragraph 1).

  1. “Line 406, please modify the way to cite the author (use the et al.).”

- “…with coauthors” is replaced with “…et al.” Through the whole text.

  1. “I recommend complementing the bibliography with an industrial case study https://doi.org/10.3390/min12040487 “

- The case is included (section “3.3. Effects of redox and temperature conditions”).

Thank you once more.  

Our best wishes for coming 2023!  

Sincerely yours,

Mikhail Vainshtein and Tatiana Abashina

Reviewer 2 Report

General comments: The present manuscript dealt with the review on modern trends in bioleaching of metals, in particular focusing on the influential role of genomics due to the recent advancements in this field. Although some part of the manuscript is well discussed in most of the review articles published time-to-time, a separate section on the associated problems with the bioleaching of arsenopyrite ore is somehow differing this manuscript with the earlier published articles. The topic is suitable for this journal, albeit, there are several points which should be addressed before its further consideration. They are: -

1.       Title: The title of the manuscript is too general which never give any idea on its main focused domain i.e., genomics. Hence, it is suggested to modify the title accordingly and also give space to “arsenopyrite” in the modified title.

2.       The same issue is with the introduction. It is suggested to build-up a research hypothesis before going to review the published articles. It should come in the last paragraph of the intro section.

3.       I’ll also suggest to build up research hypothesis and defining the novelty of this review, comparison with recently published review articles on bioleaching would be more worthy. For example, not necessarily limit to, https://doi.org/10.1016/j.jhazmat.2018.08.050.

4.       Many articles included in this review paper are very old. From my viewpoint, when the authors are discussing the recent advancements then the articles older than 5 years must not be therein. Otherwise, the manuscript will lose its relevance.

5.       It seems to me that the authors kept their manuscript limited to the bioleaching of ores only. In contrast, the recent trend of bioleaching is more inclined to the metals’ extraction form waste materials. The adaptation of microorganism is not much difficult with the ore bodies, but the high concentration of metals in waste materials with multi-elements system is more toxic to the microorganism wherein the role of modified microorganism becomes vital to study. Therefore, this reviewer would like to suggest for the inclusion of the waste materials’ bioleaching to make the manuscript more versatile and increasing the scope of the manuscript on a greater aspect. The authors may refer to https://doi.org/10.1016/j.chemosphere.2021.131978; https://doi.org/10.1039/D2GC00874B; etc.

6.       There is no research methodology on how the articles were selected and separated, what duration of the published articles are included, what topic is emphasized therein, etc. This section should come after the introduction.

7.       For a review article, the major drawing of the review article can be find in the future direction, challenges, and perspectives. They should be included as a separate section before the conclusion of the manuscript.

8.       Nevertheless, the manuscript is focused on the role of microbial genomics in bioleaching, there should be some illustrative example in the manuscript that could define the bioleaching process and the role of genomic. If possible, present some comparative example of bioleaching by using wild culture and modified microorganism that could significantly improve the process efficiency.

Author Response

Dear Reviewer,  

First of all, we want to thank for your comments and suggestions which were really helpful.  

Comments are addressed below. The responses to all comments and questions are presented below in the order of your comments.

Each comment is first recalled then the corresponding replies are given. The questions and comments of the reviewer are presented below with our answers provided as “point by point”. We corrected and supplemented the manuscript according to all these comments and questions. 

Kindly find the revised editorial article microorganisms-2096604 "Review - Current trends in metal biomining and problems of applied genomics and arsenopyrite leaching". Changes to the text are highlighted in color.  

Comments and Suggestions for Authors

Reviewer 2.

  1. “Title: The title of the manuscript is too general which never give any idea on its main focused domain i.e., genomics. Hence, it is suggested to modify the title accordingly and also give space to “arsenopyrite” in the modified title.”

- Thank you, the title is modified in accordance with this recommendation:

Review - Current trends in metal biomining. Problems of applied genomics and arsenopyrite leaching.

  1. “The same issue is with the introduction. It is suggested to build-up a research hypothesis before going to review the published articles. It should come in the last paragraph of the intro section.”

- The recommendations are implemented: the introduction was significantly expanded, its ending was modified.

  1. “I’ll also suggest to build up research hypothesis and defining the novelty of this review, comparison with recently published review articles on bioleaching would be more worthy. For example, not necessarily limit to, https://doi.org/10.1016/j.jhazmat.2018.08.050 “.

- Thank you. Page 6: “There are a number of modern good reviews on metal mining / bioleaching (where biomining is limited with natural raw materials and bioleaching include technogenic secondary sources) [1, 7, 10, 19, 21-23]. Most of them are devoted to the successes in industrial bioleaching.” Text following the quotation is some comparison and description of the review’s aim.

  1. “Many articles included in this review paper are very old. From my viewpoint, when the authors are discussing the recent advancements then the articles older than 5 years must not be therein. Otherwise, the manuscript will lose its relevance.”

- List of references is renewed: at the present time, half of the list is presented with fresh publications. Other publications are a background to show trends in the described areas.  

  1. “It seems to me that the authors kept their manuscript limited to the bioleaching of ores only. In contrast, the recent trend of bioleaching is more inclined to the metals’ extraction form waste materials. The adaptation of microorganism is not much difficult with the ore bodies, but the high concentration of metals in waste materials with multi-elements system is more toxic to the microorganism wherein the role of modified microorganism becomes vital to study. Therefore, this reviewer would like to suggest for the inclusion of the waste materials’ bioleaching to make the manuscript more versatile and increasing the scope of the manuscript on a greater aspect. The authors may refer to https://doi.org/10.1016/j.chemosphere.2021.131978; https://doi.org/10.1039/D2GC00874B; etc.”

- The review focuses on biomining (the name has been changed accordingly by recommendation of the Reviewer), anyway, additional text and references on the mentioned topic are inserted (page 2, the 2nd paragraph below Fig. 1)..

  1. “There is no research methodology on how the articles were selected and separated, what duration of the published articles are included, what topic is emphasized therein, etc. This section should come after the introduction.”

- Some additional explanations are added (page 6, paragraphs 3-4).

  1. “For a review article, the major drawing of the review article can be find in the future direction, challenges, and perspectives. They should be included as a separate section before the conclusion of the manuscript.”

- The section "6. Concluding remarks" is supplemented.

  1. “Nevertheless, the manuscript is focused on the role of microbial genomics in bioleaching, there should be some illustrative example in the manuscript that could define the bioleaching process and the role of genomic. If possible, present some comparative example of bioleaching by using wild culture and modified microorganism that could significantly improve the process efficiency.”

- Alas, at present, there are no examples of applications of genomics in industrial bioleaching. By this reason we added into the text some suggestion about possible promising direction. The revised text stipulates that the most understandable application would be to increase resistance to toxic compounds.

Thank you once more.  

Our best wishes for coming 2023!  

Sincerely yours,

Mikhail Vainshtein and Tatiana Abashina

Round 2

Reviewer 2 Report

The authors revised the manuscript well reaching up to the acceptance level.

Author Response

Dear Reviewer,  

Thank you for your kind cooperation and the great work we made together.  

The revised MS microorganisms-2096604 is submitted.

The title is changed for “Current Trends in Metal Biomining with a Focus on Genomics Aspects and Attention to Arsenopyrite Leaching - A Review”.

English is improved by professional translator.

Paragraphs “Author Contributions” and “Data Availability Statement” are supplemented.

All corrections are marked up using the "Track Changes" function and are easily viewed.

Our Season Greetings and best wishes for 2023!